# Emotional Intelligence and Perceived Social Support: Its Relationship with Subjective Well-Being

**DOI:** 10.3390/healthcare12060634

**Published:** 2024-03-12

**Authors:** Sergio Hidalgo-Fuentes, Isabel Martínez-Álvarez, María Josefa Sospedra-Baeza, Manuel Martí-Vilar, César Merino-Soto, Filiberto Toledano-Toledano

**Affiliations:** 1Department of Basic Psychology, Faculty of Psychology and Speech Therapy, Universitat de València, Av. Blasco Ibáñez, 21, 46100 Valencia, Spain or sergio.hidalgo.f@udima.es (S.H.-F.); manuel.marti-vilar@uv.es (M.M.-V.); 2Department of Psychology, Faculty of Health Science and Education, Universidad a Distancia de Madrid (UDIMA), Carretera de la Coruña Km. 38500, vía de Servicio Número 15, Collado Villalba, 28400 Madrid, Spain; 3Department of Education, Faculty of Health Science and Education, Universidad a Distancia de Madrid (UDIMA), Carretera de la Coruña Km. 38500, vía de Servicio Número 15, Collado Villalba, 28400 Madrid, Spain; isabel.martinez.al@udima.es; 4Department of Education and School Management, Faculty of Philosophy and Educational Sciences, Universitat de València, Av. Blasco Ibáñez, 30, 46100 Valencia, Spain; maria.jose.sospedra@uv.es; 5Instituto de Investigación de Psicología, Universidad de San Martín de Porres (Perú), Av. Tomás Marsano 232, Lima 34, Peru; sikayax@yahoo.com.ar; 6Unidad de Investigación en Medicina Basada en Evidencias, Hospital Infantil de México Federico Gómez, National Institute of Health, Dr. Márquez 162, Doctores, Cuauhtémoc, Mexico City 06720, Mexico; 7Unidad de Investigación Multidisciplinaria en Salud, Instituto Nacional de Rehabilitación Luis Guillermo Ibarra Ibarra, Calzada México-Xochimilco 289, Arenal de Guadalupe, Tlalpan, Mexico City 14389, Mexico; 8Dirección de Investigación y Diseminación del Conocimiento, Instituto Nacional de Ciencias e Innovación para la Formación de Comunidad Científica, INDEHUS, Periférico Sur 4860, Arenal de Guadalupe, Tlalpan, Mexico City 14389, Mexico

**Keywords:** social support, psychological well-being, happiness, emotional intelligence, life satisfaction

## Abstract

The well-being of people is a key aspect of the field of psychology. Hence, it is important to analyse the variables that are related to life satisfaction and happiness as perceived by individuals and that, therefore, increase their overall well-being. The main objective of this study was to analyse the predictive capacity of emotional intelligence and perceived social support on both the level of life satisfaction and perceived happiness. A total of 380 psychology students completed the Trait Meta Mood Scale, the Multidimensional Scale of Perceived Social Support, the Satisfaction with Life Scale, and the Subjective Happiness Scale. The results show that both emotional intelligence and social support are related to and predictive of subjective happiness and life satisfaction. The importance of developing the components of emotional intelligence and promoting an adequate social network in young people is highlighted.

## 1. Introduction

The quality of social support provided by family, teachers, or friends significantly influences the holistic development of an individual [1]. This implies that contextual factors are closely linked to psychological well-being. Additionally, emotional intelligence has been identified as an explanatory variable for individuals’ psychosocial adjustment [2].

Traditionally, psychology has focused to a greater extent on mental illness and disorder rather than on well-being to alleviate negative effects. The subjective perception of well-being received little attention within the field of psychology until the second half of the 20th century saw an increase in interest in this aspect due to its influence on people’s mental and physical health [3,4]. This boom in the study of psychological well-being arose mainly from positive psychology, a discipline that marked a new paradigm in which well-being began to be analysed from the study of variables related to it, such as happiness and life satisfaction [5]. People’s well-being includes both cognitive (life satisfaction) and affective (emotions such as happiness) aspects [6]. Research with both adolescents and adults shows that subjective well-being is related to the effective functioning of a person in various areas of his or her life and that there is a direct relationship between such well-being and positive personal qualities that favour adaptation [7,8,9,10,11].

University students often experience stress related to burnout or a low sense of self-efficacy concerning their academic tasks and responsibilities. Therefore, their emotional abilities play a crucial role in preserving their personal well-being [12]. Consequently, it is important to analyse the variables influencing their well-being and happiness to contribute to their proper evolution, both academically and personally.

Emotional intelligence is a concept that has attracted increasing interest in recent decades. Traditionally, a distinction was made between two types of models of emotional intelligence: ability models and trait models [13]. Ability models understand emotional intelligence as a cognitive ability related to the processing of emotional information [14]. Emotional intelligence in ability models is evaluated using maximum performance tests. Trait models conceptualise emotional intelligence as a set of dispositions hierarchically inferior to personality traits that determine the way in which people deal with their own emotions and the emotions of others [15]. Trait emotional intelligence is typically assessed through self-report tests. Alongside ability models and trait models are the so-called mixed models, which define emotional intelligence as a set of cognitive abilities, emotional competencies, and stable personality traits [16]. Several studies have shown a positive relationship between self-perceived emotional intelligence and personal well-being, both in its cognitive and affective components [17,18,19,20,21,22,23,24]. Likewise, meta-analytic studies have also shown a positive relationship between emotional intelligence and subjective well-being in various populations [25,26,27]. Indeed, emotional intelligence is positively associated with good mental health, leading to lower levels of anxiety and depression while enhancing self-esteem. One possible explanation for the positive relationship between emotional intelligence and subjective well-being is that emotional intelligence is also related to prosocial behaviour, a type of behaviour that not only has positive effects on those who receive help but also functions as a protective factor for the mental health and subjective well-being of those who engage in helping behaviours, also contributing to the development of communities [28]. In academia, emotional intelligence has been positively related to variables such as academic performance [29], self-efficacy [30], and academic engagement [31].

Human beings inherently possess a need to interact and thrive in society, making social relationships a focal point within the field of psychology. The interest in studying the construct of social support arises from the practicality and necessity of comprehending how these interactions between individuals unfold. Relationships with others can have positive effects on our emotional intelligence and, consequently, on personal well-being. Social relationships have the potential to alleviate tension and foster an optimal personal state. Perceived social support is defined as an individual’s perception that they have a social network to turn to in times of need [32]. Nevertheless, there is no unanimous agreement regarding its definition, so further investigation in that direction is necessary. Empirical evidence has consistently shown relationships between perceived social support and well-being, as well as quality of life [16,33,34,35,36,37,38,39,40]. Perceived social support is a crucial aspect of well-being development as it responds to the individual’s needs throughout their development [3]. Similar to emotional intelligence, perceived social support also holds significant implications in the educational realm, being associated with higher academic performance [41] and lower dropout rates [42]. Social support is considered a safeguard for both physical and psychological health and well-being. Hence, it is relevant to analyse these variables in an integrated manner. Understanding the influences among perceived social support, emotional intelligence, and physical and psychological well-being allows progress in a still underexplored path. People with adequate emotional competencies tend to engage positively in relationships and are more likely to perceive greater social support [1]. In terms of social behaviour, higher emotional intelligence contributes to a more positive self-perception of social competence. Individuals with high emotional intelligence demonstrate an enhanced ability to understand and regulate emotions, which correlates with personal well-being. Furthermore, these individuals can transfer and apply these skills to others’ emotions, thus fostering positive social relationships. Despite these insights, studies specifically analysing the relationship between social support, emotional intelligence, and personal well-being remain scarce. Moreover, findings in this area have been inconsistent. While some studies suggest that aspects like clarity and repair in emotional intelligence predict social support [43], others do not find a significant relationship between these variables [44]. In summary, although the connection between social support, emotional intelligence, and personal well-being may appear evident, only a limited number of studies have thoroughly examined the direct or mediated relationships among them. The theoretical framework we have just presented provides us with the foundations on which this research is based which stems from the need to deepen the study of the relationships between emotional and cognitive variables that influence the well-being and happiness of people, and more specifically of young people who are studying for a career and who will soon have to make crucial decisions for their lives.

Along with the positive effects of perceived well-being in the mental health domain [45,46], several studies in the educational domain have found a positive relationship between personal well-being and academic performance [47,48,49,50], an association confirmed in a meta-analysis conducted by Bücker et al. [51]. Likewise, the well-being of university students also shows a negative relationship with burnout [52,53], so studying the possible predictors of students’ well-being can play an important role in developing proposals to improve their performance and educational experience. Although, as we have pointed out above, there is previous research that has studied the relationships between emotional intelligence and social support with perceived student well-being, most of it was conducted by analysing these variables independently, especially in the Spanish population. Some studies have emphasised that perceived social support acts as a mediator in the relationship between perceived emotional intelligence and life satisfaction. According to Rey and Extremera [54], perceived emotional intelligence is positively associated with high levels of life satisfaction and perceived social support. Furthermore, this study reveals that emotional intelligence has a significant effect, both directly and indirectly (through perceived social support), on life satisfaction. The lingering question is the exploration of how emotional intelligence interacts with social support and how these factors collectively influence personal well-being. The study of factors associated with the subjective well-being of university students can provide valuable information to both educators and decision makers in order to develop interventions aimed at promoting well-being in this population. Therefore, the objectives of this study are (1) to analyse the relationships among emotional intelligence and perceived social support, life satisfaction, and subjective happiness in Spanish university students and (2) to analyse the predictive capacity of emotional intelligence and perceived social support on both life satisfaction and subjective happiness in Spanish university students. According to the theoretical framework presented above, emotional intelligence and perceived social support are expected to be positively related and predict part of the variance in the subjective well-being of university students.

Based on these objectives, a series of hypotheses is proposed and will be tested: (1) there will be a strong association between emotional intelligence and perceived social support, life satisfaction, and subjective happiness; and (2) emotional intelligence and perceived social support will contribute to explaining a portion of life satisfaction and subjective happiness.

## 2. Materials and Methods

### 2.1. Participants

To determine the sample size necessary for this study, the criteria of Green [55] were followed; these criteria propose a range of between 15 and 25 subjects for each predictor of the regression model, resulting in a minimum number of between 120 and 200 subjects, since eight variables were introduced in the regression analyses. The sample of this study, selected by incidental nonprobabilistic sampling, was finally 380 students pursuing a psychology degree at the University of Valencia aged between 18 and 55 years (M = 21.13; SD = 4.95). Of the participants, 316 were female (83.2%) and 64 were male (16.8%).

### 2.2. Instruments

Trait Meta Mood Scale (TMMS-24; Salovey et al. [55]). This self-perceived emotional intelligence scale measures through the dimensions of attention to feelings, which represents the degree to which people believe they pay attention to their own emotional states; emotional clarity, defined as the perceived ability to correctly identify and understand one’s own emotions; and emotion repair, or subjective ability to eliminate negative emotions and prolong positive ones. The Spanish adaptation [56], which has 24 items that are assessed on a five-point Likert scale, presents reliability coefficients of α = 0.86 for the emotional attention and emotion repair scales and α = 0.90 for the emotional clarity scale.

Multidimensional Scale of Perceived Social Support (MSPSS; Zimet et al. [57]). This scale evaluates the subjective perception that people have of the social support they receive in three dimensions: family, friends, and significant others. This scale is composed of 12 items that are answered on a seven-point Likert scale. The Spanish adaptation [58] of this scale showed a reliability of α = 0.85.

Satisfaction With Life Scale (SWLS; Diener et al. [59]). This scale evaluates general satisfaction with life by means of five items assessed by a seven-point Likert scale. The Spanish adaptation of this scale [60] showed a reliability of α = 0.85.

Subjective Happiness Scale (SHS; Lyubomirsky and Lepper [61]). This scale evaluates the subject’s degree of perceived happiness through four items with a seven-point Likert-type response scale. The Spanish adaptation of this scale [62] presented a reliability of α = 0.81.

### 2.3. Procedure

The study authors contacted professors from the psychology degree at the University of Valencia to obtain permission to administer the questionnaire in one of their classes. They attached an explanation of the study’s objectives as well as an example of the protocol that would be administered to the students. All students participated voluntarily and anonymously after being informed of the scope and objectives of the study. The tests were administered, after the participants provided informed consent, following the corresponding instructions for each test in the classroom during the students’ academic schedule by one of the research authors in a single session with no time limit. No incentives were offered to the students in exchange for their participation.

#### Ethical Considerations

This study is a part of a research project (HIM/2015/017/SSA.1207, “Effects of mindfulness training on psychological distress and quality of life of the family caregiver”) approved by the Research, Ethics, and Biosafety Commissions of the Hospital Infantil de México Federico Gómez National Institute of Health in Mexico City. While conducting this study, the ethical rules and considerations for research with human subjects currently enforced in Mexico [63] and those outlined by the American Psychological Association [64] were followed. All students were informed of the objectives and scope of the research and their rights according to the Declaration of Helsinki [65]. The students who agreed to participate in the study signed an informed consent form. Participation in this study was voluntary and did not involve payment.

### 2.4. Data Analysis

Data analysis was performed using the IBM SPSS Statistics v.24 statistical package. First, descriptive statistics and correlational analysis of the variables under study were calculated. Then, to determine the predicted variance in both life satisfaction and subjective happiness by emotional intelligence and perceived social support, two hierarchical regression analyses were performed. The order in which the variables were entered was the same in both cases. In the first step, the sociodemographic variables sex and age were entered; in the second step, the three scales of perceived social support (significant others, family, and friends) were entered simultaneously; and in the third step, the emotional intelligence scales (attention, clarity, and repair) were included.

## 3. Results

Means, standard deviations and Spearman correlations for the variables studied are presented in Table 1. As shown in this table, the TMMS-24 variables emotional clarity and emotion repair maintain a positive correlation with both life satisfaction and subjective happiness. The variables of perceived social support (significant others, family, and friends) also correlate positively with life satisfaction and subjective happiness. Additionally, the TMMS-24 variables show positive correlations with the three variables of perceived social support.

Two multiple hierarchical regression analyses were carried out, one for each of the dependent variables: life satisfaction (Table 2) and subjective happiness (Table 3). The results show that for both life satisfaction and subjective happiness, the models are significant when the variables of perceived social support and emotional intelligence are entered in the second and third steps, respectively. The variables sex and age explain virtually no percentage of the variance in either life satisfaction (SWLS) or subjective happiness (SHS). In step 2, the introduction of the perceived social support variables explained 24.1% of the variance in SWLS (R = 0.491, R^2^ = 0.241; F(5.374) = 23.794, *p* < 0.001) and 17.4% of the variance in SHS (R = 0.417, R^2^ = 0.174; F(5.371) = 15.749, *p* < 0.001), with perceived support from family and friends being statistically significant predictors for both dependent variables. In the last step of the regression analysis, the introduction of the three emotional intelligence variables causes the explained percentage of SWLS to increase to 40.4% (R = 0.635, R^2^ = 0.404; F(8.371) = 31.413, *p* < 0.001) and that of SHS to increase to 46.4% (R = 0.681, R^2^ = 0.464; F(8.371) = 40.135, *p* < 0.001), with the predictor variables being statistically significant for the three components of emotional intelligence for both life satisfaction and subjective happiness: attention to feelings negatively, emotional clarity and emotion repair positively.

To rule out multicollinearity, variance inflation values and tolerance indices were calculated and found to be within the recommended ranges, with variance inflation values below 10 and tolerance indices above 0.10.

## 4. Discussion

The objectives of this research were to analyse the relationships among emotional intelligence, perceived social support, subjective happiness, and life satisfaction, as well as the predictive capacity of the first two on happiness and satisfaction.

On the one hand, correlation analyses have shown the existence of strong and positive statistically significant relationships of the emotional clarity and emotion repair components of emotional intelligence with both subjective happiness and life satisfaction, results consistent with the previous findings of studies that also used the TMMS-24 test [12,18,22]. Therefore, we can say that those students who identify and understand their own emotions and who are able to manage them effectively, prolonging the positive ones and reducing the negative ones, can feel happier and more satisfied. Although emotional intelligence has generally been attributed a positive effect on life satisfaction, happiness, and mental health, the fact that most studies examining these relationships are cross-sectional hinders establishing the directionality of these relationships. However, some longitudinal studies, such as the one conducted by Dawel et al. in 2021 [66], have found a bidirectional relationship between these variables. We also found statistically significant relationships of a positive nature, although generally more moderate, of the three valued components of perceived social support (family, friends, and significant others) with perceived happiness and life satisfaction, similar to what has been found in other research papers [3,33,67,68]. Thus, there is a connection between feeling happy and satisfied and the support received from the social network formed by people in the family environment, friends, and other circles that are important to the person. Along with the above, and although it is not the main objective of the study, statistically significant relationships were also found between all the components of emotional intelligence and the components of perceived social support, a result similar to that found in previous research [1,69,70]. This result was expected since greater emotional intelligence can help increase people’s social competence and, consequently, improve their relationships and social support networks [71].

On the other hand, and consistent with previous findings [12,21,22,72,73,74], regression analyses demonstrate the importance of emotional intelligence in predicting both life satisfaction and subjective happiness: positively, the components of emotional clarity and emotion repair, and negatively, attention to feelings. Thus, students who correctly identify their emotions and are able to manage them effectively perceive themselves as happier and more satisfied. These results are consistent with previous findings that emotional clarity is a key component in predicting people’s well-being [21,75]. However, emotional mindfulness is a negative predictor of both life satisfaction and happiness. Therefore, although a certain level of emotional attention may be adaptive, an excess of attention to one’s emotions accompanied by the inability to regulate them adequately may cause a greater presence of negative states [76]. Emotional intelligence has been related to an increase in social relationships and a higher frequency of prosocial behaviours [27,77], variables that have also been associated with subjective well-being, given that maintaining frequent social relationships is a crucial aspect in the life satisfaction and happiness of the people who form part of a community.

Perceived social support, both from friends and family, is also shown to be a significant variable in predicting life satisfaction and subjective happiness. The results from previous research are congruent with this fact, although they generally relate perceived family support more strongly to personal well-being [77,78,79]. In addition to being associated with both good mental and physical health [80,81], receiving social support from multiple sources also helps people cope adaptively with life difficulties [82,83,84], so it does not seem strange that this variable functions as a predictor of personal well-being.

Despite these findings, it is necessary to consider certain limitations of this study that should be kept in mind in future research. The main limitation is that the nonprobabilistic sampling coupled with the homogeneity of the sample, composed of mostly female university students, limits the generalisability of the results obtained. Future studies would benefit from the use of random sampling. Furthermore, the wide age range of this study’s participants could also be considered a limitation, although the standard deviation of the age of the subjects is not high and most of them are in the same age group. Another limitation of this study is its cross-sectional design, since a longitudinal investigation could provide information that would help better understand the influence of emotional intelligence and social support on perceived well-being. As such, the correlation and regression analyses conducted do not allow for the establishment of mediation or moderation relationships between the variables under study. Therefore, future studies should examine such relationships, as some articles have found that perceived social support mediates the relationship between emotional intelligence and certain mental health outcomes and subjective well-being [85,86]. Finally, emotional intelligence was measured with a self-report test, so it would be useful to complement this type of measure with others of maximum performance.

## 5. Conclusions

Despite the above limitations, this study provides valuable information on the effects and predictive capacity of emotional intelligence and perceived social support on life satisfaction and subjective happiness that can be used for the design of intervention programs in various settings (school, university, family, etc.). Bearing in mind that it is possible to increase emotional intelligence with appropriate instructional programs [87,88,89], it can be concluded, therefore, that fostering emotional intelligence and reinforcing social support networks is an appropriate strategy to increase the well-being of university students, both at a general level and in specific fields of studies. The findings of the present study contribute to and extend the literature highlighting the positive associations between emotional intelligence, perceived social support, and subjective well-being.

## Figures and Tables

**Table 1 healthcare-12-00634-t001:** Descriptive analysis and bivariate correlations.

	Mean (SD)	Skewness	Kurtosis	1	2	3	4	5	6	7	8
TMMS-Attention	29.40 (5.89)	−0.53	−0.04	-							
TMMS-Clarity	25.94 (6.27)	−0.02	−0.42	0.230 **	-						
TMMS-Repair	25.95 (6.18)	−0.05	−0.65	0.044	0.425 **	-					
MSPSS-Significant others	24.63 (4.42)	−1.75	2.83	0.105 *	0.215 **	0.160 **	-				
MSPSS-Family	23.15 (5.29)	−1.35	1.31	0.135 **	0.220 **	0.250 **	0.439 **	-			
MSPSS-Friends	24.56 (4.09)	−1.66	3.37	0.109 *	0.147 **	0.279 **	0.477 **	0.389 **	-		
SWLS	25.63 (5.80)	−0.73	0.47	0.029	0.403 **	0.479 **	0.365 **	0.395 **	0.417 **	-	
SHS	4.80 (1.22)	−0.55	−0.23	−0.023	0.398 **	0.609 **	0.239 **	0.307 **	0.342 **	0.639 **	-

Notes: TMMS: Trait Meta Mood Scale; MSPSS: Multidimensional Scale of Perceived Social Support; SWLS: Satisfaction with Life Scale; SHS: Subjective Happiness Scale; * *p* < 0.05. ** *p* < 0.01.

**Table 2 healthcare-12-00634-t002:** Hierarchical multiple regression predicting life satisfaction.

	R^2^	F	B	SE	β	*p*	
Step 1	0.003	0.654					0.520
Age			0.007	0.060	0.006	0.914	
Gender			0.914	0.799	0.059	0.253	
Step 2—MSPSS	0.241	23.794					<0.001
Significant others			0.053	0.069	0.041	0.440	
Family			0.330	0.056	0.301	<0.001	
Friends			0.385	0.073	0.271	<0.001	
Step 3—TMMS	0.404	31.413					<0.001
Attention			−0.105	0.042	−0.106	0.013	
Clarity			0.207	0.044	0.224	<0.001	
Repair			0.277	0.044	0.295	<0.001	

Notes: TMMS: Trait Meta Mood Scale; MSPSS: Multidimensional Scale of Perceived Social Support. First step: R^2^ = 0.003; second step: ∆R^2^ = 0.238; third step: ∆R^2^ = 0.162.

**Table 3 healthcare-12-00634-t003:** Hierarchical multiple regression predicting subjective happiness.

	R^2^	F	B	SE	β	*p*	
Step 1	0.009	1.724					0.180
Age			0.023	0.013	0.096	0.064	
Gender			0.025	0.167	0.008	0.883	
Step 2—MSPSS	0.174	15.749					<0.001
Significant others			0.003	0.015	0.012	0.823	
Family			0.048	0.012	0.210	<0.001	
Friends			0.083	0.016	0.278	<0.001	
Step 3—TMMS	0.464	40.135					<0.001
Attention			−0.025	0.008	−0.123	0.003	
Clarity			0.037	0.009	0.189	<0.001	
Repair			0.094	0.009	0.478	<0.001	

Notes: TMMS: Trait Meta Mood Scale; MSPSS: Multidimensional Scale of Perceived Social Support. First step: R^2^ = 0.009; second step: ∆R^2^ = 0.165; third step: ∆R^2^ = 0.290.

## Data Availability

The raw data supporting the conclusions of this article will be made available by the authors without undue reservation.

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
