# Peer review of "Emotional Intelligence and Perceived Social Support: Its Relationship with Subjective Well-Being"

_healthcare, 2024, doi:10.3390/healthcare12060634_

Round 1
Reviewer 1 Report
Comments and Suggestions for Authors
Paper seems adequately referenced. Two improvements:
1. Describe incidental nonprobability sampling in depth. Sounds like the sample is a professor's class.
2. One does not know if reverse causation exists, e.g., satisfied/happy --> emotional regulation clarity (though most likely a bidirectional effect, suggesting a possible mediator and/or moderator).
Thank you for pointing out that this study is college students and primarily women. Being in a psychology class, the self-awareness is probably different as well.
Author Response
Comments and Suggestions for Authors
Paper seems adequately referenced. Two improvements:
- Describe incidental nonprobability sampling in depth. Sounds like the sample is a professor's class.
Reply: Many thanks for your contribution. In the procedure section, the collection method for the final sample has been included.
The study authors contacted professors from the Psychology degree at the University of Valencia to obtain permission to administer the questionnaire in one of their classes. They attached an explanation of the study's objectives as well as an example of the protocol that would be administered to the students
- One does not know if reverse causation exists, e.g., satisfied/happy --> emotional regulation clarity (though most likely a bidirectional effect, suggesting a possible mediator and/or moderator).
Reply: Thank you very much. The reviewer's observation is correct. Our study was not intended to contribute to this research gap, but to explore the relationships between the variables. Our results, however, still add useful evidence for the accumulation of knowledge on these variables. The potential bidirectionality between emotional intelligence and subjective well-being has been addressed in the discussion. Likewise, the study acknowledges the limitation of establishing causal relationships between variables due to the cross-sectional design employed in the study.
Although emotional intelligence has generally been attributed a positive effect on life satisfaction, happiness, and mental health, the fact that most studies examining these relationships are cross-sectional hinders establishing the directionality of these relationships. However, some longitudinal studies, such as the one conducted by Dawel et al. in 2021 [66], have found a bidirectional relationship between these variables.
As such, the correlation and regression analyses conducted do not allow for the establishment of mediation or moderation relationships between the variables under study. Therefore, future studies should examine such relationships, as some articles have found that perceived social support mediates the relationship between emotional intelligence and certain mental health outcomes and subjective well-being [85,86].
Thank you for pointing out that this study is college students and primarily women. Being in a psychology class, the self-awareness is probably different as well.
Reviewer 2 Report
Comments and Suggestions for Authors
The study described in this manuscript is yet another step in the study of EI as a potential resource that within certain circumstances and interactions may increase wellbeing and satisfaction among people in times characterized by multiple changes and crises. However - I do think the authors need to reframe their study and conduct a series of revisions for the study to offer the added value it potentially has. The following are my comments and proposed revisions:
1. English language issues are very common in the text ( see the first sentence in the abstract for a shining example). The manuscript MUST undergo language editing by a native speaker of English.
2. My main concern revolves around the study questions and hypotheses (which are currently not phrased at all): there is already ample evidence associating various measures of EI and Various reports of social support with wellbeing or its opposite - distress, anxiety and stress. To add something new the study must offer more than just associations - for example - suggest a mediation or moderation model whereas the role of EI and social support are specified and go beyond mere "there is an association between X and Y" such models will help clarify the roles of EI and social support and the inter-relations among them. For such examples see:
Zysberg, L., & Zisberg, A. (2022). Days of worry: Emotional intelligence and social support mediate worry in the COVID-19 pandemic. Journal of Health Psychology, 27(2), 268-277.
as well as
Kornas-Biela, D., Martynowska, K., & Zysberg, L. (2023). ‘With a Little Help from My Friends’: Emotional Intelligence, Social Support, and Distress during the COVID-19 Outbreak. International Journal of Environmental Research and Public Health, 20(3), 2515.
Such studies have already suggested models that go beyond the association pattern, and may serve as reference to pose your hypotheses and suggested models against.
3. In the introduction concepts are presented but the connections between them and why were they selected for this study is unclear. The text must be revised and restructured to build a rationale for a model tested in this study. How do you expect EI to interact with social support and how are both concepts associated with wellbeing , satisfaction or happiness? once you present a conceptual framework - the literature to review becomes much clearer and should be adjusted accordingly. The authors must make it clear - what is missing in the available body of knowledge about EI, SS and wellbeing/ happiness and how they set out to propose a way to address that lacuna. I propose they end the section with a set of well defined hypotheses, or a graphic model to present the interrelations proposed by the study.
4. The result analysis will have to change based on my suggested revision above: Correlations and regressions will not be enough. Path analyses or at lease stepwise regressions (if we are testing a moderation effect) will be required here.
5. The discussion may be revised based on the above changes - what does the current study model and results add to existing knowledge of the concepts at the focus of the text? what is new? Suggestions for future studies need to be raised especially in light of the results and the study limitations.
Comments on the Quality of English LanguageSee my comments above
Author Response
Comments and Suggestions for Authors
The study described in this manuscript is yet another step in the study of EI as a potential resource that within certain circumstances and interactions may increase wellbeing and satisfaction among people in times characterized by multiple changes and crises. However - I do think the authors need to reframe their study and conduct a series of revisions for the study to offer the added value it potentially has. The following are my comments and proposed revisions:
- English language issues are very common in the text ( see the first sentence in the abstract for a shining example). The manuscript MUST undergo language editing by a native speaker of English.
Reply: Thank you very much for your feedback. The English writing of the manuscript will be reviewed by a professional translator.
- My main concern revolves around the study questions and hypotheses (which are currently not phrased at all): there is already ample evidence associating various measures of EI and Various reports of social support with wellbeing or its opposite - distress, anxiety and stress. To add something new the study must offer more than just associations - for example - suggest a mediation or moderation model whereas the role of EI and social support are specified and go beyond mere "there is an association between X and Y" such models will help clarify the roles of EI and social support and the inter-relations among them. For such examples see:
Zysberg, L., & Zisberg, A. (2022). Days of worry: Emotional intelligence and social support mediate worry in the COVID-19 pandemic. Journal of Health Psychology, 27(2), 268-277.
as well as
Kornas-Biela, D., Martynowska, K., & Zysberg, L. (2023). ‘With a Little Help from My Friends’: Emotional Intelligence, Social Support, and Distress during the COVID-19 Outbreak. International Journal of Environmental Research and Public Health, 20(3), 2515.
Such studies have already suggested models that go beyond the association pattern, and may serve as reference to pose your hypotheses and suggested models against.
Reply: Many thanks for your contribution. The hypotheses have been incorporated.
Based on these objectives, a series of hypotheses is proposed and will be tested: (1) there will be a strong association between emotional intelligence and perceived social support, life satisfaction, and subjective happiness; and (2) emotional intelligence and perceived social support will contribute to explaining a portion of life satisfaction and subjective happiness.
The new analyses suggested by the reviewer go beyond the objectives of the present study, they have been included as potential avenues for future exploration.
As such, the correlation and regression analyses conducted do not allow for the establishment of mediation or moderation relationships between the variables under study. Therefore, future studies should examine such relationships, as some articles have found that perceived social support mediates the relationship between emotional intelligence and certain mental health outcomes and subjective well-being [85,86].
85.Kornas-Biela, D.; Martynowska, K.; Zysberg, L. ‘With a Little Help from My Friends’: Emotional Intelligence, Social Support, and Distress during the COVID-19 Outbreak. Int. J. Environ. Res. Public Health, 2023, 20, 2515.
86.Zysberg, L.; Zisberg, A. Days of worry: Emotional intelligence and social support mediate worry in the COVID-19 pandemic. J. Health Psychol. 2020, 1–10.
- In the introduction concepts are presented but the connections between them and why were they selected for this study is unclear. The text must be revised and restructured to build a rationale for a model tested in this study. How do you expect EI to interact with social support and how are both concepts associated with wellbeing , satisfaction or happiness? once you present a conceptual framework - the literature to review becomes much clearer and should be adjusted accordingly. The authors must make it clear - what is missing in the available body of knowledge about EI, SS and wellbeing/ happiness and how they set out to propose a way to address that lacuna. I propose they end the section with a set of well defined hypotheses, or a graphic model to present the interrelations proposed by the study.
Reply: Thank you very much. The introduction has undergone substantial expansion, placing particular emphasis on elucidating the relationships among the study variables. An in-depth explanation of the connections between the selected variables is provided to clarify the rationale behind their inclusion in this study. These variables were meticulously chosen based on their theoretical relevance and potential impact on the research objectives. Each variable is expected to contribute valuable insights into the relationships and patterns under investigation, thereby bolstering the overall robustness and comprehensiveness of the study.
The quality of social support provided by family, teachers, or friends significantly influences the holistic development of an individual [1]. This implies that contextual factors are closely linked to psychological well-being. Additionally, emotional intelligence has been identified as an explanatory variable for individuals' psychosocial adjustment [2].
University students often experience stress related to burnout or a low sense of self-efficacy concerning their academic tasks and responsibilities. Therefore, their emotional abilities play a crucial role in preserving their personal well-being [12]. Consequently, it is important to analyze the variables influencing their well-being and happiness to contribute to their proper evolution, both academically and personally.
Indeed, emotional intelligence is positively associated with good mental health, leading to lower levels of anxiety and depression while enhancing self-esteem.
Human beings inherently possess a need to interact and thrive in society, making social relationships a focal point within the field of Psychology. The interest in studying the construct of social support arises from the practicality and necessity of comprehending how these interactions between individuals unfold. Relationships with others can have positive effects on our emotional intelligence and, consequently, on personal well-being. Social relationships have the potential to alleviate tension and foster an optimal personal state. Perceived social support is defined as an individual's perception that they have a social network to turn to in times of need [32]. Nevertheless, there is no unanimous agreement regarding its definition, so further investigation in that direction is necessary. Empirical evidence has consistently shown relationships between perceived social support and well-being, as well as quality of life [16,33–40]. Perceived social support is a crucial aspect of well-being development as it responds to the individual's needs throughout their development [3]. Similar to emotional intelligence, perceived social support also holds significant implications in the educational realm, being associated with higher academic performance [41] and lower dropout rates [42]. Social support is considered a safeguard for both physical and psychological health and well-being. Hence, it is relevant to analyze these variables in an integrated manner. Understanding the influences among perceived social support, emotional intelligence, and physical and psychological well-being allows progress in a still underexplored path. People with adequate emotional competencies tend to engage positively in relationships and are more likely to perceive greater social support [1]. In terms of social behavior, higher emotional intelligence contributes to a more positive self-perception of social competence. Individuals with high emotional intelligence demonstrate an enhanced ability to understand and regulate emotions, which correlates with personal well-being. Furthermore, these individuals can transfer and apply these skills to others' emotions, thus fostering positive social relationships. Despite these insights, studies specifically analyzing the relationship between social support, emotional intelligence, and personal well-being remain scarce. Moreover, findings in this area have been inconsistent. While some studies suggest that aspects like clarity and repair in emotional intelligence predict social support [43], others do not find a significant relationship between these variables [44]. In summary, although the connection between social support, emotional intelligence, and personal well-being may appear evident, only a limited number of studies have thoroughly examined the direct or mediated relationships among them.
Some studies have emphasized that perceived social support acts as a mediator in the relationship between perceived emotional intelligence and life satisfaction. According to Rey and Extremera [54], perceived emotional intelligence is positively associated with high levels of life satisfaction and perceived social support. Furthermore, this study reveals that emotional intelligence has a significant effect, both directly and indirectly (through perceived social support), on life satisfaction. The lingering question is the exploration of how emotional intelligence interacts with social support and how these factors collectively influence personal well-being.
- The result analysis will have to change based on my suggested revision above: Correlations and regressions will not be enough. Path analyses or at lease stepwise regressions (if we are testing a moderation effect) will be required here.
Reply: Thank you very much for your feedback. As previously discussed, the additional analyses suggested by the reviewer extend beyond the objectives of the present study. Nevertheless, they have been included as potential avenues for future research
- The discussion may be revised based on the above changes - what does the current study model and results add to existing knowledge of the concepts at the focus of the text? what is new? Suggestions for future studies need to be raised especially in light of the results and the study limitations.
Reply: Thank you very much. The reviewer's observation is correct. As the reviewer rightly points out, future lines of research have been incorporated, taking into account both the limitations of the study and the obtained results.
As such, the correlation and regression analyses conducted do not allow for the establishment of mediation or moderation relationships between the variables under study. Therefore, future studies should examine such relationships, as some articles have found that perceived social support mediates the relationship between emotional intelligence and certain mental health outcomes and subjective well-being [85,86].
85.Kornas-Biela, D.; Martynowska, K.; Zysberg, L. ‘With a Little Help from My Friends’: Emotional Intelligence, Social Support, and Distress during the COVID-19 Outbreak. Int. J. Environ. Res. Public Health, 2023, 20, 2515.
86.Zysberg, L.; Zisberg, A. Days of worry: Emotional intelligence and social support mediate worry in the COVID-19 pandemic. J. Health Psychol. 2020, 1–10.
Reviewer 3 Report
Comments and Suggestions for Authors
My suggestions are: 1) tighten up the literature review; 2) strengthen the coherence; and 3) undertake a proofreading
For example, the following paragraphs needs to be improved to make it clearer.
Perceived social support has been defined as an individual’s perception that he or she has a social network to turn to in case of need [29]. Empirical evidence has repeatedly shown relationships between perceived social support and well-being and quality of life [14,30–37]. Perceived social support is an essential aspect of the development of well-being because it responds to the needs presented by the individual during his or her development [1]. As in the case of emotional intelligence, perceived social support also has important implications in the educational sphere, being related to higher academic performance [38] and to lower dropout rates [39].
Comments on the Quality of English LanguageMy suggestions are: 1) tighten up the literature review; 2) strengthen the coherence; and 3) undertake a proofreading
For example, the following paragraphs needs to be improved to make it clearer.
Perceived social support has been defined as an individual’s perception that he or she has a social network to turn to in case of need [29]. Empirical evidence has repeatedly shown relationships between perceived social support and well-being and quality of life [14,30–37]. Perceived social support is an essential aspect of the development of well-being because it responds to the needs presented by the individual during his or her development [1]. As in the case of emotional intelligence, perceived social support also has important implications in the educational sphere, being related to higher academic performance [38] and to lower dropout rates [39].
Author Response
Comments and Suggestions for Authors
My suggestions are: 1) tighten up the literature review; 2) strengthen the coherence; and 3) undertake a proofreading
For example, the following paragraphs needs to be improved to make it clearer.
Perceived social support has been defined as an individual’s perception that he or she has a social network to turn to in case of need [29]. Empirical evidence has repeatedly shown relationships between perceived social support and well-being and quality of life [14,30–37]. Perceived social support is an essential aspect of the development of well-being because it responds to the needs presented by the individual during his or her development [1]. As in the case of emotional intelligence, perceived social support also has important implications in the educational sphere, being related to higher academic performance [38] and to lower dropout rates [39].
Comments on the Quality of English Language
My suggestions are: 1) tighten up the literature review; 2) strengthen the coherence; and 3) undertake a proofreading
For example, the following paragraphs needs to be improved to make it clearer.
Perceived social support has been defined as an individual’s perception that he or she has a social network to turn to in case of need [29]. Empirical evidence has repeatedly shown relationships between perceived social support and well-being and quality of life [14,30–37]. Perceived social support is an essential aspect of the development of well-being because it responds to the needs presented by the individual during his or her development [1]. As in the case of emotional intelligence, perceived social support also has important implications in the educational sphere, being related to higher academic performance [38] and to lower dropout rates [39].
Reply: Thank you very much for your feedback The paragraph highlighted by the reviewer has been carefully reviewed and corrected, with additional review by a professional translator planned for the entire text. Furthermore, the introduction has been strengthened to improve coherence. The elucidation of the relationships between variables has been reinforced, along with the reinforcement of the theoretical framework supporting the study.
Human beings inherently possess a need to interact and thrive in society, making social relationships a focal point within the field of Psychology. The interest in studying the construct of social support arises from the practicality and necessity of comprehending how these interactions between individuals unfold. Relationships with others can have positive effects on our emotional intelligence and, consequently, on personal well-being. Social relationships have the potential to alleviate tension and foster an optimal personal state. Perceived social support is defined as an individual's perception that they have a social network to turn to in times of need [32]. Nevertheless, there is no unanimous agreement regarding its definition, so further investigation in that direction is necessary. Empirical evidence has consistently shown relationships between perceived social support and well-being, as well as quality of life [16,33–40]. Perceived social support is a crucial aspect of well-being development as it responds to the individual's needs throughout their development [3]. Similar to emotional intelligence, perceived social support also holds significant implications in the educational realm, being associated with higher academic performance [41] and lower dropout rates [42]. Social support is considered a safeguard for both physical and psychological health and well-being. Hence, it is relevant to analyze these variables in an integrated manner. Understanding the influences among perceived social support, emotional intelligence, and physical and psychological well-being allows progress in a still underexplored path. People with adequate emotional competencies tend to engage positively in relationships and are more likely to perceive greater social support [1]. In terms of social behavior, higher emotional intelligence contributes to a more positive self-perception of social competence. Individuals with high emotional intelligence demonstrate an enhanced ability to understand and regulate emotions, which correlates with personal well-being. Furthermore, these individuals can transfer and apply these skills to others' emotions, thus fostering positive social relationships. Despite these insights, studies specifically analyzing the relationship between social support, emotional intelligence, and personal well-being remain scarce. Moreover, findings in this area have been inconsistent. While some studies suggest that aspects like clarity and repair in emotional intelligence predict social support [43], others do not find a significant relationship between these variables [44]. In summary, although the connection between social support, emotional intelligence, and personal well-being may appear evident, only a limited number of studies have thoroughly examined the direct or mediated relationships among them.